# Caspase-3 Mediated Cell Death in the Normal Development of the Mammalian Cerebellum

**DOI:** 10.3390/ijms19123999

**Published:** 2018-12-12

**Authors:** Laura Lossi, Claudia Castagna, Adalberto Merighi

**Affiliations:** Department of Veterinary Sciences, University of Turin, Grugliasco (TO) I-10095, Italy; claudia.castagna@unito.it (C.C.); adalberto.merighi@unito.it (A.M.)

**Keywords:** caspase-3, cell death, apoptosis, cell proliferation, neurons, glia, cerebellum, development

## Abstract

Caspase-3, onto which there is a convergence of the intrinsic and extrinsic apoptotic pathways, is the main executioner of apoptosis. We here review the current literature on the intervention of the protease in the execution of naturally occurring neuronal death (NOND) during cerebellar development. We will consider data on the most common altricial species (rat, mouse and rabbit), as well as humans. Among the different types of neurons and glia in cerebellum, there is ample evidence for an intervention of caspase-3 in the regulation of NOND of the post-mitotic cerebellar granule cells (CGCs) and Purkinje neurons, as a consequence of failure to establish proper synaptic contacts with target (secondary cell death). It seems possible that the GABAergic interneurons also undergo a similar type of secondary cell death, but the intervention of caspase-3 in this case still remains to be clarified in full. Remarkably, CGCs also undergo primary cell death at the precursor/pre-migratory stage of differentiation, in this instance without the intervention of caspase-3. Glial cells, as well, undergo a process of regulated cell death, but it seems possible that expression of caspase-3, at least in the Bergmann glia, is related to differentiation rather than death.

## 1. Introduction

Caspase-3 is a key enzyme in the execution of apoptosis. The enzyme belongs to the caspase (from cysteine-aspartate protease) family, one of the six families of proteases that have important functions in normal neuronal development and neuropathology [1]. Caspases are endoproteases and mediate some forms of neuronal programmed cell death (PCD), such as apoptosis and pyroptosis, which are dependent on caspase-3 and caspase-1, respectively [2].

During neurodevelopment, a process of PCD occurs by which neurons that are produced in excess are removed from tissue to sculpt the mature brain and spinal cord. PCD may depend or not on the activation of caspases and thus can be didactically subdivided into a caspase-dependent and -independent PCD. The former consists of caspase-3-dependent apoptosis and caspase-1-dependent pyroptosis. Today there are no reports on the occurrence of pyroptosis in cerebellum, whereas, as it will be discussed in this review, caspase-3 has long been known to be involved into cerebellar PCD. Caspase-independent PCD consists of calpain-dependent autophagy and receptor-interacting serine/threonine-protein kinases 1, 3 and mixed lineage kinase domain-like pseudokinase (RIPK1-RIPK3-MLKL) -dependent necroptosis, a process which is limited by cathepsins through cleavage of RIPK1 (see [1] for a very recent review). Although some forms of autophagic cell death were observed during normal postnatal cerebellar development in normal mice [3] as well as in *Reeler* and *Lurcher* mutants [3,4], it appeared that, at least in *Lurcher*, cell death was not caused by autophagy but was rather a necrosis with autophagic features, both in vivo and in vitro [4]. In addition, it was recently demonstrated that the administration of calpeptin, a potent, cell-permeable calpain inhibitor, promoted the apoptotic cell death of the cerebellar granule cells (CGCs) in postnatal rats [5], and that mutations in the gene encoding for calpain-1 in several mammals, including humans, resulted in alterations of cerebellar development, with increased apoptotic cell death of the CGCs and ataxia [6]. Therefore, it seems highly probable that calpain is protective against the apoptotic cell death of CGCs during development, rather than being involved in the autophagy of these neurons.

To date, necroptosis has been described only in the cerebellum of Niemann-Pick type C1 (NPC1) diseased mice and human patients, primarily affecting the Purkinje neurons (PNs) [7,8]. Therefore, among the several types of PCD that occur during cerebellar development caspase 3-dependent apoptosis is the most physiologically relevant along the process of naturally occurring (physiological) neuronal death (NOND). In the past, we reviewed the cellular and molecular mechanisms of neuronal apoptosis in the mammalian central nervous system (CNS) [9]. We subsequently demonstrated the occurrence of two different cellular mechanisms of cerebellar NOND in rabbit [10] and mouse [11]. In the two species, we showed that apoptosis of proliferating precursors and young pre-migratory cells occurred in the absence of caspase-3 cleavage, whereas the death of post-mitotic, post-migratory neurons was directly linked to caspase-3 activation. We here provide an up-to-date picture of the intervention of the protease in the normal course of mammalian cerebellar development. There are certain discrepancies in the investigations that have, until now, been dedicated to this subject, and we will try to highlight these from a correct and critical perspective.

Readers who are not familiar with the basic molecular and cellular mechanisms of neuronal cell death would much benefit from referring to one of the numerous existing reviews on these subjects, as we will only mention some relevant issues to put discussion under the right frame of reference here. To be better cognizant of the significance (or, at times, insignificance) of the research that we have here described, one should also, crucially, be aware of the technical drawbacks that are associated with a proper identification of dead and proliferating cells, since, as we will discuss widely in the following sections, during cerebellar development, the two processes are tightly interconnected.

## 2. The Recognition of Cell Death in Cerebellar Neurons

### 2.1. The Struggle for the Recognition of Cell Death of Neurons

It has been written that the tyranny over evidence is nowhere more glaringly apparent than in the history of the delayed discovery of neuronal death during normal development [12]. At the basis of such a “tyranny”, there was the theory that both ontogeny and phylogeny were progressive, from lower and less organized nervous systems to higher and more organized brains. Evidence of NOND was indeed provided as early as the 1880s [13,14], but it was not acknowledged, because it conflicted with the widely accepted notion of progressive development. Therefore, reports of neuronal death were interred in the scientific literature, to be unburied much later as queer archeological remains. Today, these papers come back to obsess us, as they obsessed previous generations of neuroscientists, who could not accept any knowledge that went against their treasured theories.

The long delay in believing the existence of developmental neuronal death has been regarded as an historical enigma, as discussed in [12]. Deterministic theories of development made it hard to conceive of the demolition of temporary structures as a part of the normal development, as it was not easy to imagine, at that time, that construction and destruction could be proceeding simultaneously during ontogeny.

The first descriptions of cell death in neurons were provided in fish and amphibians by Beard [13,14], who, at the end of the nineteenth century, described the loss of a specific population of mechanoreceptors in the spinal cord of these animals. A few years later, in 1906, Collin reported the occurrence of dorsal root ganglion and motor neuron death during the embryonic development of chicks [15]. After two decades, Ernst first realized that in several areas of the CNS and peripheral nervous system, an overproduction of neurons was indeed followed by the death of a substantial fraction of them [16]. He was also the first to propose a general theory of developmental neuron death, to obtain supporting evidence for his theory, and to describe the existence of three main types of cell death during normal neurodevelopment: the first, afterwards named *phylogenetic* by Glücksmann [17], occurring during the regression of vestigial organs; the second, then referred to as *morphogenetic*, during the morphological modifications of organ anlages; the third, subsequently called *histogenetic*, during tissue remodeling.

Glücksmann’s 1951 review of cell death during normal development was indeed the first publication to prime discussion on the importance of neuronal cell death in experimental limb amputation [17], a widely diffused experimental paradigm in neuroembryology of the time. A few years later, Hamburger was able to show that it was an increased cell death, and not a mitotic or a differentiation failure, to be responsible for the decrease in the number of motorneurons in the chick embryo after limb amputation [18]. Then, Hughes showed the occurrence of a large overproduction of motorneurons during normal development in *Xenopus* and that the death of these cells was a critical factor for the control of their final number [19,20], and Prestige demonstrated the same phenomenon in tadpole dorsal root ganglion and spinal cord neurons [21].

Years later, in the 1970s, the idea that tissue assembly could take place by means of a selective deletion of temporary structures started to gain more attention. Before that time, the dominant idea, boosted by the cogent evidence in support of the neurotrophic theory, established after the original work of Levi-Montalcini and Cohen [22], was that matching between the number of central neurons and their peripheral targets was physiologically achieved by co-orchestrated programs of cell proliferation, migration, and differentiation, but *not* death. This explains why works on developmental neuronal death initially received very little attention. However, after these pioneering studies, the idea that a form of PCD, already theorized conceptually, indeed occurred at different stages of development and growth of the nervous system and that PCD was independent of external insults quickly began to take hold. Eventually, PCD was recognized as a highly phylogenetically conserved physiological mechanism by which eukaryotic cells die following a stereotyped series of molecular and cellular events during the development and differentiation of all tissues and organs, including those of the nervous system.

### 2.2. The Discovery of PCD and NOND in Cerebellum

In altricial mammals, i.e., mammals with inept offspring, a substantial fraction of cerebellar development occurs postnatally, within a temporal window of about two-three weeks in rodents and rabbits and up to three months in humans. At these stages, the cerebellar cortex is made of four layers named, from the most superficial contacting the pia mater, the external granular layer (EGL), the molecular layer, the Purkinje cells’ layer and the internal granular layer (IGL). The EGL is temporary, and disappears upon completion of cerebellar differentiation. Thus, only three layers remain in the mature cortex, where the IGL becomes the single granular layer to be recognized [23]. Acknowledgement of cerebellar NOND inside this temporal window had also to face the dogmatic views of its time. Therefore, it was not surprising that the first report on the subject, dating back to the end of the 1970s, concluded that death did not contribute significantly to cerebellar development [24]. In those years, the initial studies on cerebellar cell death were focused onto the alterations induced by thyroid dysfunction during postnatal development, rather than on normal cerebellar neurogenesis [25,26]. One of these two studies considered, by the use of quantitative autoradiography, the effects of experimental hypo- and hyperthyroidism on the rates of cell proliferation and generation in the EGL of P10 rats, reporting that both conditions reduced the mitotic rate [25], albeit with different mechanisms. Namely, hyperthyroidism was referred to shorten the cell cycle by decreasing the duration of the G_1_ phase and it was hypothesized that thyroid over-functioning promoted an early neuronal differentiation of the EGL precursors that stopped to proliferate and acquired the typical features of maturity. On the other hand, hypothyroidism had no effect onto the length of the cell cycle, but reduced the rate of cell acquisition in the absence of any significant change in cell death [25]. Hypothyroidism was, instead, described to be responsible of an increased cell death in the IGL of two- to three-week postnatal rats, with a peak at P10 in the other mentioned study [26]. Notably, the authors reported that, at this stage of development, cell death was maximal also in normal control animals. A few years later, in the early 1980s, signs of cell degeneration were described in the form of pyknotic cells in normal P5, P10 and P12 hamster cerebella; these pyknotic nuclei were by far more evident in the deep half of the IGL that, at all the three postnatal ages studied, showed a greater cell loss than the superficial half [27]. A series of papers then followed, in which cell death was described in several cerebellar mouse ataxic spontaneous mutants including *Lurcher* [28], *Pcd* [29], *Staggerer* [30,31], *Nervous* [32] and *Weaver* [33].

In 1993, Wood and colleagues published a seminal paper demonstrating that cell death in the postnatal mouse cerebellum was apoptotic [34]. To do so they used the terminal deoxynucleotidyl transferase dUTP nick end labeling (TUNEL) technique, an in situ DNA end labeling procedure for detecting the double-strand DNA breaks typical of apoptosis [35]. The application of the TUNEL method (and other in situ DNA labeling techniques) to the study of cell death, as well as the use of different biochemical approaches to detect oligonucleosomal DNA fragments in tissue extracts allowed demonstrating that cerebellar NOND was apoptotic in several mammals, including all the most commonly used laboratory species and humans [36]. Qualitative (agarose gel electrophoresis and TUNEL) and quantitative (ELISA for oligonucleosomal DNA fragments) estimates of apoptosis demonstrated that the rat cerebellum displayed an initial intense apoptotic peak at P10 and a second smaller peak at P21 [37]. The P10 peak was coincident with the timing of cell death reported in earlier purely morphological studies on neurodegeneration that were mainly based onto the observation of nuclear pyknosis. In parallel, apoptosis was reported to be the main form of cell death occurring in *Lurcher* [38] and *Weaver* [39] mutants and to affect the cerebellar granule cells when the PNs were ablated by, e.g., the specific expression of diphtheria toxin in transgenic mice [40]. At the same time, another study used the TUNEL method and electron microscopy to confirm that the pyknotic cells in the postnatal cerebellar cortex were indeed undergoing apoptosis and, very importantly, employed immunocytochemistry to better characterize the nature of the dying cells in the white matter and IGL, reaching the conclusion that 50–70% of the pyknotic cells were astrocytes [41].

## 3. Caspase-3 in Cerebellar Development

### 3.1. Modes of Cell Death in Neurons and Their Relevance to Cerebellar Development

It is now widely accepted that a substantial fraction of the neurons produced during neurogenesis die before the full maturation of the nervous system, and that nearly all classes of neurons are produced in excess during development. These oversized populations of neurons are then significantly reduced during NOND, by a relatively limited number of widely investigated modalities that are not exclusive to neurons but, in a more general context, affect all eukaryotic cells [42].

The importance of morphological observation for a correct identification of the mode of cell death in neurons (and other cells) is rooted into the classic discrimination between apoptosis—the first morphological form of PCD to be recognized (see below)—and necrosis, a term that was originally employed by Virchow to indicate the death of cells in response to a pathological event [43]. After PCD was acknowledged as a common phenomenon in development, an ultrastructural study on several embryonic tissues proposed that there were three main types of cell death, on the basis of the role played by lysosomes in cell disruption [44]. In the first type, death occurs in the absence of an activation of the cell’s endogenous lysosomes. Nonetheless, cells are eventually disposed of by tissue macrophages’ phagocytosis and secondary lysosome activation. This process has also been more precisely referred to as *heterophagocytosis*. In *autophagocytosis*, cells are instead removed from tissues after activation of their own lysosomal enzymes, whereas in the third type of cell death, there is no obvious lysosome intervention. The first two types are by far the more common and have been described by various authors starting from the 1960s. The ultrastructural features of type 1 cell death in Schweichel and Merker classification [44] correspond to the current definition of apoptosis, while at least a variant of type 3 shares several features with necrosis [45].

Today several other kinds of cell death have been recognized in neurons [46], each displaying a typical cohort of structural and biomolecular changes, but with widespread occurrence of intermediate features that make the entire array of the forms that cell death can assume a continuum rather than constituting fully separate entities (Figure 1). Among the several configurations of cell death that can occur in NOND, apoptosis is the one that, by a considerable margin, most directly relates to the present discussion.

#### Apoptosis

Although often considered the prototype form of PCD—and often erroneously indicated as PCD—apoptosis is only one of several types of NOND. Such a misinterpretation derived from the initial recognition of apoptosis as the most widely diffused mode of neuronal cell death.

Apoptosis was originally defined as a distinct type of cell death according to a series of characteristic ultrastructural features that depend on a stereotyped sequence of events starting from nuclear and cytoplasmic condensation, and leading to cell fragmentation and, eventually, phagocytosis [47]. Initially Kerr and co-authors [47] used the term *shrinkage necrosis* to describe this form of cell death. Subsequently, they coined the word apoptosis (from the Greek = falling of the leaves), which indicates the dropping of leaves from trees or petals from flowers, to emphasize its occurrence in normal tissue turnover.

Apoptosis involves a chain of morphologically well-defined steps that can only in part be appreciated by light microscopy, but are most clear at the transmission electron microscope (TEM) level (Figure 1).

Nuclear changes are the first unequivocal sign of apoptosis: at the start, they consist in chromatin condensation and segregation into very electron-dense, sharply delineated masses that abut the nuclear envelope. High-magnification TEM images reveal that these masses of chromatin are made up of closely packed, fine granular material. Initial condensation of the nucleus eventually leads to true nuclear pyknosis. At the same time, cytoplasmic condensation also occurs, with an increase in the density of the cytosol and the appearance of some vacuoles. The cell organelles retain a healthy appearance but become closely packed, likely because of a remarkable cytosolic loss. Ribosomes get separated from the rough endoplasmic reticulum and polysomes, to eventually disappear. As apoptosis proceeds further, the cell and its nucleus become more irregular in shape: the nucleus buds to produce discrete fragments, still surrounded by an intact envelope. The subsequent ultrastructural steps in nuclear and chromatin condensation are paralleled by a well-characterized sequence of molecular events. Eventually, the cell fragments into membrane-bounded apoptotic bodies that are cleared from tissue by the resident microglia [48].

Apoptosis is accompanied by a specific pattern of DNA damage that leads to the formation of low-molecular weight DNA oligomers. These can be visualized after DNA isolation and electrophoresis in tissue extracts, or directly in situ by the use of the TUNEL technique [49]. The other main molecular features of apoptosis are the release of cytochrome c from mitochondria, and the activation of caspases, including caspase-3, in the cytosol [50,51]. It is well accepted that two major pathways can be differentiated by considering the relative timing of caspase activation and mitochondrial release of cytochrome c during apoptosis. In the first, which is exemplified by activation of death receptors, an effector caspase is activated prior to mitochondrial alterations (*extrinsic pathway*). In the second, cytochrome c is released from the mitochondrial intermembrane space prior to caspase activation (*intrinsic pathway*). Notably, caspase-3 is the most important effector caspase upon which both pathways converge. Molecules along the mitochondrial pathways comprise the proteins of the B-cell lymphoma 2 (BCL-2) family, which, in turn, is composed of pro- and anti-apoptotic members, the molecules of the apoptosome, among which are caspase-9, the apoptotic peptidase activating factor 1 (Apaf1) and the cytochrome c, and the family of the inhibitors of apoptosis proteins (IAPs).

Another molecule of interest in the present context is the apoptosis-inducing factor (Aif) protein. Aif is localized into mitochondria, between the two mitochondrial membranes, and is thus separated from the cell nucleus. However, when mitochondria are damaged, Aif moves into the cytosol and thence to the nucleus. Here, Aif triggers chromatin condensation and DNA fragmentation to induce PCD independently from caspase activation (see [9] for original references).

### 3.2. Intervention of Caspase-3 in Cerebellar Development

#### 3.2.1. Basic Features of Cerebellar Development

During cerebellar development, well-orchestrated sequences of proliferative, differentiative, migratory and death processes occur, and they affect the different populations of cerebellar neurons and glia (Figure 2).

The cerebellar anlage displays two primary and two secondary proliferative zones (see [23] for a recent review). The primary proliferative zones are the ventricular zone (VZ), i.e., the neuroepithelium lining the walls of the fourth ventricle, and the rhombic lip (RL); the secondary proliferative zones are the external germinal zone (EGZ) and the rostral germinal zone (RGZ). The VZ gives rise to all GABAergic cerebellar neurons, including the PNs and the interneurons [52,53]. The RL is part of the neural fold of the dorsal edge of the fourth ventricle. Its upper part has long been known to generate the glutamatergic cerebellar granule cells (CGCs) [54]. The CGC precursors are generated between E15.5 to E17 in rat and mouse and migrate in an outward direction to originate the EGZ. During the first postnatal weeks, the EGZ forms the EGL where the CGCs are ultimately generated. More recent studies have demonstrated that, at earlier developmental stages (E9–E13.5 in mice), the EGZ also generates the glutamatergic neurons of the cerebellar nuclei and the unipolar brush cells. The EGZ additionally gives birth to part of the Golgi cells, at least in certain species. Finally, the RGZ has a specific role in the development of the rostral cerebellum. After proliferation has ceased, post-mitotic neurons migrate to their final destinations within the cerebellar cortex or the cerebellar nuclei. As we will discuss below, cell death may hit the different subpopulations of cerebellar neurons at their pre- or post-mitotic stages of differentiation. Death, for the most part, takes the form of apoptosis, and may or not be strictly related to caspase-3 activation.

#### 3.2.2. CGCs

CGCs are the most abundant neurons in the cerebellum and by far the most widely studied cerebellar neurons in the context of cell death. The EGL where they are generated can be subdivided further into an outer and an inner portion, respectively populated by highly proliferating precursors and non-proliferating, post-mitotic pre-migratory CGCs. As mentioned (see Section 3.2.1), the time window in which the generation of these neurons occurs is strictly related to the degree of maturation of the cerebellum at birth. In altricial mammals, the window spans from the first week after birth to a few weeks (rodents, rabbits) or months (humans) thenceforth. Soon after they are generated, CGCs migrate in a downward direction along the Bergmann glia to reach the IGL, which will become their final location, as the only granular layer in the mature cerebellar cortex. In parallel with their maturation, CGCs undergo impressive morphological changes and massive cell death. Thus, the generation of CGCs relies on a controlled balance between clonal expansion and NOND. The two processes are tightly interconnected in time and space, but still poorly understood with regard to their causative links. Studies on these matters have been carried out in humans, as well as in several laboratory species.

##### 3.2.2.1. Human Studies

In human fetuses, one of the first surveys on cerebellar development used classical histology and proliferating cell nuclear antigen (PCNA) immunostaining to evaluate cell proliferation, and TUNEL to label the nuclei with fragmented DNA and, hence, the occurrence of apoptosis. This study was carried out between the 16th and the 24th week of gestation and reported that, on topographic bases, there was not a mutual relationship between the two events [55]. In a later investigation, where cell proliferation was examined between the 24th week and the first postnatal year using Ki-67 immunostaining, it was observed that, in the EGL, proliferation mainly occurred between the 28th and the 34th gestational week and persisted until the 8th postnatal month, whereas, in IGL, Ki-67 labeled cells were detected only between the 24th and the 36th week of gestation [56]. In the same paper, the occurrence of very low percentages (0.3% at maximum) of degenerating pyknotic cells was reported, as well. Afterwards, human cerebella from the 17th week of gestation to one year postnatally were studied semi-quantitatively, again using PCNA immunostaining and TUNEL [57]. It was thus shown that PCNA positive cells were present in all layers of the cerebellar cortex from the 17th to the 25th prenatal week. In the EGL, proliferating cells persisted at high levels up to the 39th week, declining thereafter and disappearing three months postnatally. TUNEL-positive cells were reported within the EGL only, beginning from the 26th prenatal week and persisting up to one year after birth. Notably, it was observed that these cells did not appear to suffer morphologically and were negative for BCL-2 [57]. Another study on embryonic and fetal material (from the 8th to the 22nd week of gestation) used the TUNEL method, as well as propidium iodide to stain the dead cells, and found very high percentages of positive cells in the germinal zones of the cerebellar primordium, up to about 35% in the VZ and more than 30% in the EGZ, but only about 5% of the TUNEL-positive presumably apoptotic cells were immunoreactive for the cell death receptor Fas (APO-1/CD95) [58]. This latter observation suggested that the main contribution to NOND of the CGCs, which, as mentioned above, are surely the most quantitatively relevant population of neurons originating from the EGZ, came from activation of the intrinsic apoptotic pathway. In keeping with this possibility, in an earlier study BCL-X, a member of the BCL-2 family of proteins, was reported to be highly expressed in the fetal cerebellum [59].

Other observations on the relationship between proliferation and NOND in the postnatal period, from birth to adulthood, were carried out in our laboratory by using Ki-67 immunostaining and the TUNEL and T4 DNA ligase methods [36]. The latter was employed to confirm the true occurrence of DNA fragmentation in the tissues under study, taking into consideration the problems that are linked to the use of TUNEL on materials collected at necropsy, under non-flawless conditions of tissue preservation. We thus observed that about one third of the cells in the EGL were Ki-67 positive from birth to two months postnatally, with a subsequent decline of immunoreactivity up to the end of the third month and a complete disappearance at P120. Apoptotic cells were 4–8% in the EGL, less than 2% in the IGL and less than 3% in the white matter, from birth to the third postnatal month, and thereafter disappeared. They were often immunoreactive for caspase-1, then known as human interleukin-1β-converting enzyme [60] and caspase-3, which, at that time, was called 32-kDa putative cysteine protease (CPP32) [61] (see [62] for additional information on nomenclature/classification of caspases) but did not stain for BCL-2. Notably, 16.10% of the apoptotic cells (TUNEL-positive) were immunoreactive for vimentin, a marker of the CGC precursors.

In summary, most of the studies on the development of the human cerebellum converged to demonstrate that: (1) Proliferation of CGCs spans quite a long temporal window that roughly encompasses the last two-thirds of gestation and ceases at the third–fourth postnatal month, in parallel with the disappearance of the EGL; (2) NOND is restricted to two narrower temporal windows, the first occurring during embryonic and fetal development (8–22 weeks) and the second during the first three months after birth; (3) NOND is associated with DNA fragmentation and, thus, is very likely of the apoptotic type and related to the activation of the intrinsic pathway; (4) Postnatal NOND is primarily affecting the CGC precursors and may be associated with caspase-3 activation (see detection of active caspase-3).

##### 3.2.2.2. Animal Studies

Animal studies on the intervention of caspase-3 in the apoptotic processes affecting the CGCs are substantially in line with the observations in humans. However, the possibility to experimentally label the DNA-synthesizing (proliferating) cells with ^3^H-thymidine, BrdU or other thymidine analogues has made it possible to better analyze the relationship between proliferation and NOND. Also, studies on spontaneous mutants and transgenic models have much helped to clarify some of the main cellular mechanisms that regulate NOND.

###### Relationship of cell proliferation and NOND

Not surprisingly, the initial descriptions of proliferating or dying cells during cerebellar development relied upon the observation of the cellular morphology with the detection of mitoses, cytoplasm fragmentation or nuclear condensation, and pyknosis (for a recent review see [23]). Animal studies have been focused on the postnatal period, as the most commonly used laboratory species (rat, mouse, rabbit) are altricial.

###### Rats

In rats, the adult granular layer is fully mature around P50 [23]. In these studies, both the mitotic index and the percentage of degenerating cells, mainly calculated in the EGL, were quite low (less than 3%) [23]. As mentioned, administration of ^3^H-thymidine to label the DNA-synthesizing cells was pivotal to a better comprehension of the relationship between cell proliferation and NOND of the CGCs. Thus, a study on undernourished P9 rats, where cell death was detected on purely morphological grounds, concluded, on a purely speculative basis, that commitment to death occurred during the S-phase of the cell cycle [63]. Again on partly speculative grounds, after DNA-labeling with ^14^C-thymidine administered to P7–16 rats, death was estimated to affect less than 5% of the whole population of newly generated cells [24]. Subsequent studies, for the most, used the TUNEL method to label the apoptotic cells in the P3–16 cerebellum of normal rats [64], rats with hypothyroidism [65] or lacking the endothelin receptor B [66] and, in some instances, to correlate death with cell proliferation. In one of these studies, 54% of apoptotic EGL cells were reported to express PCNA, whereas only slightly more than 22% and 15% of apoptotic neurons were located, respectively, in the post-mitotic and pre-migratory zones of the EGL [64]. The authors thus concluded that, in EGL, dividing CGCs undergo apoptosis more frequently than post-mitotic CGCs and hypothesized the existence of developmental stage-specific mechanisms of apoptosis of these neurons. Observations on the mutant Flathead (*Fh*) rat, confirmed the close link between apoptosis and proliferation during CGCs differentiation and maturation. *Fh* rats display a flattened cranium, resting tremor, ataxia, progressive paralysis of the hind limbs, and die between P21–P28. Homozygous rats have a dramatically reduced brain size caused by a sudden and violent onset of apoptosis at E16, with a peak at E18. Late-developing structures such as the cerebellar IGL become severely depleted of cells, in parallel with the appearance of multinucleated cells in the EGL. These observations thus suggested that the aberrant cell death in *Fh* was due to a defective cytokinesis in progenitor cells [67].

Further corroboration on the close link between cell proliferation and death of the CGCs came from Western blot studies on the cerebellum of hyperbilirubinemic Gunn rats [68]. These animals exhibited an increased cell cycle arrest in the late G_0_/G_1_ phase, characterized by a decrease in the expression of cyclins A and A1 and cyclin-dependent kinase 2, which are active during the S phase and peak at the S-G_2_ transition, and cyclin D1, which governs the transition from G_1_ to the S phase and counteracts apoptosis. In parallel, the Gunn rats displayed marked increases of cyclin E, which also regulates the G_1_-S transition but amplifies the apoptotic pathway, active caspase-3 and cleaved poly ADP-ribose polymerase-1 (PARP-1), one of the main substrates of caspase-3.

###### Mice

In an early study comparing the development of the P0-P9 cerebellum in normal and *weaver* mice, mitotic figures and pyknotic cells in the EGL were always very few in normal animals (2% and less than 0.5%, respectively), mainly located in the pre-migratory half of the EGL and, in *Weaver*, were quickly cleared from tissue in four-seven hours [33]. Later, using the TUNEL method and/or after direct TEM observation, several reports have been published that converged to demonstrate the presence of variable amounts of apoptotic CGCs in the P0–10 EGL of normal mice [11,69] and mutants such as *Lurcher* [70], *Reeler* [3,71], and *Weaver* [70]. However, few investigations were devoted to analyzing the relationship of CGC proliferation and NOND in this species. In one of these studies, which employed BrdU and TUNEL labeling, active caspase-3 immunostaining and transmission electron microscopy, a quite convincing demonstration was provided that the EGL neurons died through a process of PCD soon after they exit from the cell cycle [72].

The relationship between proliferation and apoptosis is so strict that, in several circumstances, it has been hypothesized that apoptosis was initiated by a re-enter into the cell cycle. This is the case, e.g., of the *Harlequin* (*Hq*) mutant mice. The Hq mutation is a proviral insertion in the apoptosis-inducing factor gene (*aif*) that gives rise to a reduction of about 80% in Aif expression. *Hq* mice display a progressive degeneration of terminally differentiated cerebellar neurons that is the consequence of an exaggerated response to oxidative stress [73]. When *aif* was deleted at a very early stage of development using an engrailed-1 (*En1*) promoter-driven *cre* recombinase gene, a dramatic reduction in the population of CGC precursors was observed, as their proliferation was blocked before initiation of the S phase [74]. Further analysis revealed that if *aif* was deleted later in embryogenesis, using this time a nestin promoter-driven *cre* recombinase gene, CGCs survive for a longer period, but are still unable to enter the S phase [74].

We have, more recently, studied cell proliferation and NOND in the normal and *Reeler* mouse after BrdU administration and TUNEL labeling in the P0-25 interval [71]. When the density of the BrdU-positive cells in EGL was correlated with the developmental stage, the correlation curve in normal mice showed a maximum at P9 and a minimum at P25, whereas in the IGL there was a maximum at P0 followed by negligible oscillations between P15–25. Correlation curves between the numbers of apoptotic cells and the developmental stage were also calculated. In the EGL, the curve was oscillatory, with a vertical positive asymptote immediately before birth, maxima at P9.5 and P23 (this latter, however, being quite low), and minima at P3 and P19.5. In the IGL, instead, the curve was bell-shaped, starting with a vertical negative asymptote immediately after birth and a single maximum at P4.5. It is noteworthy that in the EGL, there was a temporal coincidence, around P9, of the proliferative and apoptotic peaks. This is fully in line with the results that we obtained previously in rabbits using a totally different approach, which demonstrated a very tight temporal connection (1–1.5 days) between the generation and death of precursors/pre-migratory CGCs (see Rabbits). Remarkably, *Reeler* mice displayed statistically significant differences in the extent of cell proliferation and apoptosis, as well as a derangement in their temporal trends along the timespan considered in our study. In mutants, we also employed predictive statistics to show that proliferation, rather than apoptosis, was the most important factor to determine the cellularity of the cerebellar cortex.

###### Rabbits

We analyzed the relationship of cell proliferation and death in the P0–30 rabbit cerebellum after time-window labeling of proliferating cells in vivo to study the cell cycle kinetics; we also used a variety of approaches to detect apoptosis, among which were histology and TEM analysis, immunocytochemistry, TUNEL, genomic DNA electrophoresis, Western and Southern blotting [48]. We concentrated our analysis at P5, where, in Southern blots, DNA fragmentation was maximal, as was cleavage of PARP-1 [75] in Western blots. We showed that CGCs and their precursors in the EGL went through apoptosis within 24–36 h after their generation, to be thereafter cleared from tissue by the resident microglia, an intervention that was also recently confirmed in rats [76]. After Southern blotting and immunostaining, BrdU, the first thymidine analogue administered to rabbits in the time window-procedure, was present at the level of low-size DNA oligomers as soon as twelve hours after cell division. Therefore, this apoptotic phase was intrinsic to the EGL neurons and independent of their synaptic interactions with targets. Apoptotic CGCs, although fewer in number, could also be detected in the IGL; considering the results of cell cycle kinetics measurements and the rapid tissue clearance by glia, we excluded that these neurons underwent DNA fragmentation in the EGL to subsequently migrate to their final destination, but rather postulated that post-migratory CGCs endured NOND as a failure to establish their synapses with the PNs. In a subsequent paper, we established that pre-migratory CGCs in the EGL underwent apoptosis upon phosphorylation of checkpoint kinase 1 and hyper-phosphorylation of the retinoblastoma protein, suggesting that NOND of pre-migratory CGCs was linked to activation of DNA checkpoint and alteration of the normal cell cycle parameters [10]. In parallel, we also demonstrated that apoptosis of post-migratory CGCs in the IGL was, instead, caspase-dependent (see below). Therefore, our observations in rabbits were fully consistent with earlier studies on postnatal *Lurcher* mouse mutants and wild-type/*Lurcher* chimeras where, after ^3^H-thymidine labeling of DNA-synthesizing (proliferating) cells and quantitative estimations of the CGC population and size of their synaptic target, the existence of two distinct mechanisms for developmental NOND of the CGCs were postulated [77].

###### Detection of active caspase-3

Assessment of the role of caspase-3 during CGC development in vivo was difficult for several reasons, among which were the availability of good antibodies that specifically recognized the cleaved (active) form of the enzyme, the very quick removal of the apoptotic cells from tissue, and the recently demonstrated constitutive activity of the enzyme, at least in cerebellar slices [78].

As caspase-3 is ubiquitous, the first issue is of paramount importance, and it is very critical that immunocytochemical localizations are accompanied by Western blots to assess the specificity of the primary antibodies for the cleaved fragment of caspase-3 and to establish its presence in tissue extracts. The inactive full-length caspase-3 is localized in the cytosol where it is cleaved by upstream caspases along the intrinsic pathway; then the cleaved active enzyme rapidly translocates into the nucleus of the cells undergoing PCD. As we have discussed in a previous review [9], it is notable that numerous studies reported no immunostaining for the inactive zymogen, whereas it appeared that the expression of the caspase-3 mRNA was highly regulated in parallel with the occurrence of postnatal apoptosis. CNS alterations in caspase-3-deficient mice reflected the failure of NOND during neurogenesis. In the cerebellum, but also in several other areas of CNS, these alterations, which were associated with premature lethality, consisted of a diffuse hyperplasia, presence of ectopic cell masses, abnormal structural organization and neuronal hypercellularity. Specifically, the decrease or suppression of cell death in these mice has been linked with a block of the mitochondrial apoptotic pathway.

Although caspase-3 immunoreactive cells were observed in all the species considered in the section Relationship of cell proliferation and NOND, their numbers were, in general, very low. For example, only 28% of the TUNEL-positive neurons in the postnatal human cerebellum were immunoreactive for caspase-3, so that these double-labeled cells represented about 1.1–2.2% of the EGL cells and only 0.5% of those in IGL [36]. Similarly, in non-quantitative studies, only rare active caspase-3 immunoreactive cells were reported in the rat [79], mouse [11,80] or rabbit [10] postnatal EGL or IGL, whereas another study recorded a maximum of 5.9% immunoreactive cells in EGL at P8 and 2.4% in the IGL at P9 in mice [81]. Although active caspase-3/TUNEL double-stained CGCs were observed in humans [36] and the expression of active caspase-3 was associated with positive immunostaining of IGL neurons for active caspase-7 and -9 in rabbits [10], studies in rat were controversial. An early study (which was however carried out in organotypic cerebellar slices) showed that the EGL neurons underwent apoptosis when they expressed PCNA, but not TAG-1, a marker protein of the post-mitotic and pre-migratory CGCs [82]. They were thus interpreted to be precursors/pre-migratory CGCs, and their apoptotic death was blocked by a caspase-1 inhibitor. In keeping with these ex vivo observations, it was later demonstrated that the genetic ablation of caspase-3 in transgenic mice completely inhibited NOND in EGL neurons [72]. However, another paper, instead, supported a non-apoptotic role for active caspase-3 in the EGL cells and suggested a possible involvement of the enzyme in proliferation and differentiation rather than in NOND [79]—see also Section 4. Non-Apoptotic Role of Caspase-3.

As mentioned, the immunocytochemical localization of active caspase-3 is associated with a series of potential drawbacks. To overcome these obstacles, we employed alternative approaches, based on the transfection of postnatal cerebellar slices with probes that enabled to monitor the activity of the enzyme in live CGCs. Initially, we used the ApoAlert™ pcaspase3-sensor vector to analyze the cleavage of caspase-3 during NOND [11]. This approach relied on the possibility of observing the nuclear translocation of a fluorescently tagged cytoplasmic reporter molecule after cleavage by the protease in the transfected cells. It was thus possible to confirm the existence of caspase-3 dependent and independent apoptotic phenomena affecting the CGCs at different stages of their maturation and differentiation. However, such an approach was not amenable to quantitative studies, and thus was of limited value for further pharmacological characterization. To overcome this important limitation, more recently we used a caspase-3-sensitive fluorescence resonance energy transfer (FRET) probe (pSCAT3-DEVD), which enabled us to image the active caspase ex vivo and to analyze its localization and cellular dynamics in CGCs [78]. We thus measured the ratio of the emissions of the donor/acceptor FRET pair (enhanced cyan fluorescent protein (ECFP)_em_/Venus_em_) in alive or fixed slices of the P5 mouse cerebellum. The specificity of the probe was confirmed with RNA interference and after inhibition of the protease with Ac-DEVD-CMK. It was thus possible to demonstrate that caspase-3 was constitutively active in CGCs and to identify the specific cellular compartment(s) of enzyme activation. In addition, we could follow the fluctuations of the ECFP_em_/Venus_em_ ratio along time, and its response to 25 mM KCl-induced depolarization, or to increased intracellular Ca^2+^ after *N*-methyl-d-aspartate (NMDA −1 mM), kainic acid (1 mM), or A23187 (100–200 μM).

The observation that, in postnatal CGCs, the activity of caspase-3 was dynamically regulated and did not follow an on-off switch is important because, methodologically, it highlights the necessity of a quantitative analysis of the levels of activation of the enzyme and of a critical reappraisal of the histological observations simply made by the use of antibodies against cleaved caspase-3. In addition, and in a more biologically relevant context, it gives support to the idea that there may be numerous different opposing pathways directly or indirectly converging onto caspase-3 to regulate NOND in CGCs. One of these mechanisms, which appeared to be specific for CGCs, e.g., involved the immunoglobulin superfamily adhesion molecule close homolog of L1 (CHL1) that plays important roles during the development of the nervous system. Recent work has, in fact, demonstrated that the in vivo ablation of CHL1 during the second postnatal week of mouse cerebellar development caused an enhanced death of CGCs, but not of the PNs [83]. Remarkably, immunostaining for active caspase-3 in CHL1-deficient mice increased in the IGL, but not the EGL, confirming the occurrence of a caspase-3 dependent cell death only in the former.

###### Molecular pathways and regulation of NOND in CGCs

The cellular and molecular mechanisms of NOND in the CGCs that meet onto the effector caspase-3 have been widely investigated. As expected, many molecules that intervene in the intrinsic and extrinsic apoptotic pathways in other neurons and/or non-neural cells have been localized in CGCs and directly implicated in the regulation of death in these neurons. Along the intrinsic pathway there is, however, a remarkable exception as Bax, one of the molecules of the Bcl-2 family of proteins, did not appear to be involved in the process. However, it is widely accepted today that both Bax and Bak (and in some cases, Bok) work together to induce intrinsic apoptosis, as demonstrated from observations on *bax^−/−^*-*bak^−/−^* double knockout mice that display an anomalous cell accumulation in the CNS (see Table 3 in [9] for references to the original studies on transgenic mice with a neuronal death/survival phenotype). Therefore, one cannot exclude that the observations discussed below may somewhat have underestimated the intervention of Bax in the CGC apoptosis. In adult *bax^−/−^* mice, which display a numerical reduction of the PNs (see Section 3.2.3, below), the number of CGCs was unaffected, indicating that Bax was not involved in NOND of the precursors/pre-migratory CGCs [84,85]. However, Bax seemed to regulate the post-mitotic, post-migratory CGC target-related secondary death, as more CGCs survived in *bax^−/−^*:*Lurcher* double mutants (chimeras)—where the CGC target is reduced as a consequence of the primary death of the PNs—than in control *Lurcher* mutants [84]. These observations gave further support to the notion that CGCs suffer two differently regulated types of PCD, primary and secondary, which are, respectively, target independent or dependent. Backing to this concept also came from investigations in vitro. In this experimental context, treatment of the rat CGCs with the phosphatase inhibitor, okadaic acid (OKA) or the excitatory neurotransmitter l-glutamate resulted in an apoptotic progressive death [86]. However, the OKA-induced neurotoxicity was accompanied by the activation of caspase-3, with the appearance of the typical oligonucleosomal DNA ladder in Southern blots, whereas l-glutamate-induced neurotoxicity was not accompanied by the activation of caspase-3 and other upstream caspases, or DNA laddering. Notably, both OKA and l-glutamate induced a similar pattern of nuclear DNA disintegration into high-molecular weight fragments. In addition, Z-DEVD-FMK, a specific caspase-3 inhibitor, and Z-VAD-FMK, a general caspase inhibitor, inhibited both the oligonucleosomal and high-molecular weight DNA fragmentation after OKA, but the two inhibitors were inefficacious to prevent the formation of high-molecular weight DNA fragments after l-glutamate. Based on these results, it was concluded that the formation of the high-molecular weight DNA fragments in CGCs accompanied both the caspase-dependent and -independent types of cell death, further reinforcing the concept that multiple mechanisms in the regulation of the excision of DNA loop domains occur during the death of these neurons.

Caspase-dependent PCD of the CGCs may not only follow the activation of the mitochondrial pathways, as at least some of the molecules along the death receptor pathway have also been implicated in the death of CGCs. One of these molecules is the Fas ligand (FasL), which displayed a pro-apoptotic activity, eventually leading to a reduction in thickness of the IGL, and induced a marked increase of caspase-3 activity in colorimetric assays and of immunoreactive CGCs in the IGL, bur not EGL [87]. The intervention of Fas in triggering certain forms of CGC apoptosis was also confirmed by more recent in vitro and slice studies where the anti-apoptotic protein lifeguard (LFG), an inhibitor of the death receptor pathway, was demonstrated to promote the survival of CGCs by interfering with caspase-8 activation and thence downstream caspases [88].

That both the intrinsic and extrinsic apoptotic pathways are activated in CGCs in response to different apoptotic stimuli and/or their degree of differentiation was also supported by several studies on some molecules of the IAP family, a group of structurally related proteins that are negative regulators of the activity of caspases and comprises, among others survivin, XIAP and cIAPs [89]. In cancer cells, survivin may be recruited to mitochondria and thence released back in the cytosol in response to cell death stimuli. Survivin released from the mitochondria then forms a complex with XIAP that thus becomes more stable and protects cells against proteasomal degradation, eventually leading to a synergistic inhibition of effector and initiator caspases [89]. Studies in vitro into non-neuronal cells long ago showed that survivin was capable of inhibiting caspase activity, so that apoptosis induced by Fas or Bax and executed by caspase-3 was blocked [90]. In parallel, in addition to caspase inhibition, several other functions have been ascribed to survivin, which is now unanimously recognized as an indispensable regulator of cell division and to handle the balance between cell proliferation and death [91]. Studies in which the Cre-loxP-system was used to generate mice lacking *survivin* in neuronal precursor cells [92] then demonstrated that newborn conditional mutants displayed a marked reduction in the size of the brain, associated with severe, multifocal apoptosis of many areas of CNS, among which the cerebellum. In the latter, caspase-3 and caspase-9 activities were significantly elevated, whereas, remarkably, *bax* expression was unchanged from controls. More recent observations of cultured CGCs and histological sections have localized survivin, XIAP, cIAP-1 and cIAP-2 during the course of postnatal maturation of rat cerebellum and detected an interaction between XIAP and active caspase-3 [93,94]. Even more recently, we used postnatal mouse cerebellar slices to study the functional interactions of caspase-3 and survivin in the process of NOND and experimental cell death of CGCs [78]. We thus demonstrated that survivin reduced the basal levels of caspase-3 activation in a series of double transfection experiments with the pSCAT3-DEVD caspase-3 sensor probe and a second probe, pHcRed1-C1-survivin, driving the overexpression of the protein in CGCs under the control of the HCVM promoter.

Along the two main pathways of apoptosis, among the molecules that were shown to modulate postnatal CGC survival in vivo, there is l-glutamate, as the activation or inhibition of some of its receptors, i.e., the metabotropic glutamate receptor 5 (mGluR5) [95] or the ionotropic NMDA receptor [96], resulted in an increase or a decrease of the apoptotic cell death of the CGCs in the EGL and/or IGL. Specifically, it was observed that glutamate acted on mGluR5 as a functional switch to regulate CGC survival in the EGL. Therefore, when the receptor is antagonized, NOND is reduced, and vice versa, thereby controlling the final number of CGCs in the mature cerebellum [95]. On the contrary, when NMDA receptors were blocked, particularly in IGL, increased levels of DNA fragmentation (both in Southern blots and after TUNEL) and caspase-3 (and caspase-1 at lower extent) were observed, in parallel with a decrease of ornithine decarboxylase, a marker of CGC normal differentiation [96].

Other molecules of interest are the high-affinity tropomyosin receptor kinase B (trkB) neurotrophin receptor and the low-affinity p75 neurotrophin receptor (p75^NTR^). The brain-derived neurotrophic factor (BDNF) is produced by the CGCs residing in the outer proliferative part of the EGL [9] and positively regulates the survival and migration of these neurons by binding the trkB receptors that are also expressed in EGL, very likely in an autocrine fashion, and IGL [97]. The BDNF precursor molecule (proBDNF) is, instead, pro-apoptotic. Remarkably, in the rat cerebellum between E20 and P8, the levels of the mature neurotrophin, as well as the BDNF/proBDNF ratio, negatively correlated with the expression of active caspase-3 [98].

The functions of p75^NTR^ have long been elusive, as they depend on the cell context and binding to different co-receptors to activate specific intracellular pathways. Although the positive modulation of apoptosis after brain injury is the best characterized function of p75^NTR^ [99], in the recent past it was shown that the receptor regulates the proper cell cycle exit of the CGC progenitors in the developing rat and mouse EGL, an action prompted after binding with pro-neurotrophin 3, as, in the absence of p75^NTR^, progenitors continue to proliferate beyond their normal period [100]. Even more recently, it was observed that ablation of p75^NTR^ did not directly affect NOND in mouse CGCs [101]. However, the deletion of the downstream effector receptor interacting protein 2 (RIP2) increased CGC apoptosis. Remarkably, CGC’s NOND was restored to basal levels when p75^NTR^ was deleted in RIP2-deficient mice indicating RIP2 as a crucial molecule for normal p75^NTR^ signaling in these neurons.

##### 3.2.2.3. Summary of Current Knowledge

In brief, current knowledge of the occurrence and regulation of NOND in CGCs of altricial species can be summarized as follows: (1) In all species studied so far, including humans, there are two postnatal apoptotic waves of NOND, in temporal sequence. The first wave hits the precursors and pre-migratory CGCs in outer EGL (primary apoptosis), while the second affects the mature CGCs during their migration to the IGL (secondary apoptosis). These two waves are preceded by a phase of late embryonic apoptotic cell death that has been more extensively investigated in humans; (2) Primary apoptosis may be tightly linked to cell cycle dysregulation, is independent from connection with synaptic targets and very likely may be completed without the intervention of caspase-3, although this issue might be worthy of further investigation, as the protease is constitutively and dynamically active in CGCs. If so, Aif can be the effector molecule to trigger apoptosis in the absence of caspase-3 activation. (3) Secondary apoptosis follows a failure to connect to the PN dendritic tree, the main synaptic target of the mature CGC axons, i.e., the parallel fibers; and is tightly associated with the activation of caspase-3; (4) Both primary and secondary apoptotic death of CGCs may be activated following the intrinsic or extrinsic apoptotic pathways and differentially regulated by several biologically relevant molecules. Obviously, molecules that are part of these pathways, such as survivin, play important roles in the regulation of CGCs’ NOND, with the remarkable exception of Bax. Important regulators are also the excitatory transmitter L-glutamate and BDNF with their related receptors.

#### 3.2.3. PNs

In mouse, the PNs are generated between E10–E13 in the VZ [102]. They then migrate in an outward direction along the radial glia to form a plate of immature cells in the intermediate part of the cerebellar anlage. During subsequent maturation, the plate, which was originally composed of several layers of cells, becomes a monolayer of well aligned cells, a process that is completed within the first week after birth in rat and mouse and governed by Reelin, a glycoprotein of the extracellular matrix [103] produced by CGCs [104]. NOND affects the PNs in the course of their generation and migration with a plateau during the first postnatal week and is important for late compartmentation and wiring of the cerebellar cortex [105,106,107,108]. On ultrastructural bases, NOND was apoptotic [3], depended upon caspase-3 activation as, e.g., 5.9% neurons in the Purkinje cell layer were immunostained using an antibody that recognized the cleaved form of the enzyme at P3 [81], and affected a higher number of PNs at P4 than P7 [109]. However, after combined ^3^H-thymidine autoradiography, TUNEL and active caspase-3 immunostaining it was clear that the PCD of the PNs was independent from their birth-date, i.e., it was not related to interactions with targets [109]. Immunocytochemical studies on the expression of the apoptosis-regulatory proteins Bid, Bcl-2, Bcl-X_L_, Bax and Bak during development of murine nervous system were supportive about the apoptotic nature of postnatal NOND of the PNs, as these neurons expressed high levels of Bid (pro-apoptotic) and Bcl-X_L_ (anti-apoptotic) at P7 [110]. Studies in humans are rare and limited to the embryonic period. Human PNs started being recognizable around the 12th week of gestation and became organized into a monolayer between the 16th and 28th week [111]. Initial observations on histopathological samples from 12–24 weeks of gestation [55] or 17 weeks of gestation to 12 months postnatal [57] did not report staining for PN apoptotic cells after TUNEL. However, in a subsequent investigation at 15–22 weeks of gestation, about 4% of the cells in the Purkinje cell layer were reported to be TUNEL-positive and about 5% immunoreactive for Fas, one of the main death receptors leading to PCD [58].

Much of our current knowledge of NOND of the PNs comes from studies on normal rodents to which one has to add several ataxic spontaneous mutants, see, e.g., Table 2 in [3] and [112], and numerous strains of transgenic mice. Among mouse mutants, PNs’ apoptosis has been demonstrated in *Reeler* [3,71], *Lurcher* [113,114], *Pcd* [115] and *Toppler* [116] mice, whereas in *Hotfoot* (and after experimental deletion of the ionotropic glutamate receptor delta 2—GluRδ2) delayed PN cell death, and reductions in the extension of the PN dendritic tree were observed [117].

Broadly speaking, studies on NOND in PNs from transgenic mice have: (1) helped to confirm its apoptotic nature; (2) elucidated some of the molecular and cellular mechanisms governing the process; and (3) shed some light on its biological role. Under these perspectives, many observations (discussed below) first converged to demonstrate the activation of the intrinsic mitochondrial apoptotic pathways during PNs’ NOND. Notably, when one of the better-characterized pro-apoptotic genes of the *bcl-2* family, *bax*, was deleted, there was a drastic reduction in the number of TUNEL-positive cells in the marginal zone of the cerebellar anlage. Remarkably, this is the zone where, during embryonic development, the immature PNs are found along their route of outward migration to the final position in the cerebellar cortex [118]. Therefore, it was not surprising that a misplacement of these neurons followed postnatally [85]. After TUNEL and activated caspase-3 immunostaining, the extent of NOND was drastically reduced in the Purkinje cell layer of the cerebellar cortex also in the P5 cerebellum of *bax*-KO mice [118]. Data on the intervention of Bcl-2 onto PNs’ NOND are more difficult to organize into a coherent frame. Not surprisingly, the number of PNs increased by over 30% in one line of Bcl-2 overexpressing transgenics [84]. However, when transgenic mice were generated in which the expression of human BCL-2, one of the most important negative regulator of apoptosis, or the anti-apoptotic adenovirus E1B 19k was targeted to the PNs by specific regulatory elements from the *pcp2* gene some puzzling results were obtained in another study. In this survey, amazingly, no significant numerical differences in the strongest BCL-2 expressing line were observed compared to wild-type mice, and only a 14.2% increase was noted in the pcp2/E1B 19k transgenic line [119]. Interestingly, the overexpression of *bcl-2* or the deletion of *bax* in heterozygous *Lurcher* mice only delayed the autonomous death of the PNs, which are known to undergo apoptosis as a direct consequence of the gain of function mutation of the GluRδ2 [84]. Aif appears to be another important regulator of the generation of the PNs [74]. It was in fact demonstrated that, when the encoding gene was deleted at a very early stage of cerebellar development, there was a dramatic reduction in the number of the PNs. Such a reduction was consequent to premature entrance in the S-phase of the cell cycle of the PN precursors, most of which failed to undergo mitosis to eventually die by apoptosis.

More recent observations also implicated the extrinsic death receptor apoptotic pathways in PN NOND. Thus, LFG was shown to be highly expressed during early differentiation of cerebellum and to affect cerebellar size, PN development, and IGL thickness [88]. In adult mice, LFG additionally promoted PN maintenance. This was demonstrated by observing the onset of morphological alterations and reduction in the density of the PNs, as well as increased active caspase-3 and caspase-8 immunostaining in knockout mice or mice where the expression of LFG was reduced by RNA silencing [88]. In these animals, LFG inhibited Fas-mediated cell death by interfering with caspase-8 activation; such a mechanism was supported by the detection of higher numbers of active caspase-8-positive PNs in adult mice lacking LFG.

Understanding the biological relevance of PNs’ NOND during cerebellar development is made difficult also considering the mutual supportive function with the CGCs. Nonetheless, it is relevant mentioning here that, when NOND was blocked, an enhanced PN survival compromised the normal cerebellar function, as demonstrated after behavioral experiments [119]. It has also been documented that, during normal cerebellar development, the PNs undergoing apoptotic NOND in the vermis were highly localized at the level of the midline and in a lobule specific, parasagittal pattern so that PCD and PCD-free zones could be easily observed and quantified, an observation considered to be important for cerebellar histogenesis [108]. Finally, the misplacement of PNs that occurred when *bax* was knocked out also led to the hypothesis that the removal of ectopically located PNs may be an important function of developmental PCD [85].

#### 3.2.4. GABAergic Interneurons

NOND of the cerebellar GABAergic interneurons has been little investigated. In mouse, the stellate, basket and Golgi cells are generated postnatally from the VZ [23] in a cortical area adjacent to the white matter. In monkeys, the proliferative territory of these neurons is instead close to the EGZ [120]. Migration and PCD of the stellate and basket cells during the first three postnatal weeks was studied in glutamic acid decarboxylase 67 (GAD67)/green fluorescent protein (GFP) knock-in mice where the GABAergic neurons are tagged by GFP [121]. The GFP-positive cells underwent a progressive numerical reduction from P5 to P21 in the white matter, in parallel with an increase in the number of the fluorescent neurons in the molecular layer until P15. After tracing the S-phase cells with BrdU, dual-labeled BrdU/GFP positive neurons were found in the white matter during the P2–P8 interval, in the IGL at P7, and in the molecular layer at P9 [121]. These results confirmed that the proliferation of the mouse stellate and basket cells started in the white matter and that these neurons then migrated in an outward direction to reach the molecular layer. Migration was then followed by nuclear condensation, revealed by Hoechst 33258, and an apoptotic process between P16–P17, as demonstrated after immunocytochemical staining for single strand DNA and cleaved caspase-3, and by the TUNEL method [121]. It thus seems possible that apoptosis of the GABAergic interneurons is secondary to the failure to properly connect synaptically with their target, but further observations would be required to fully clarify this point.

### 3.3. Glia

As often appears to be the case, apoptosis was comparatively poorly investigated in glia with respect to in neurons. A study on the P7 rat cerebellum reported that about 50% of the pyknotic cells in the white matter contained fragmented DNA, i.e., they were TUNEL-positive, and were immunoreactive for the astrocytic marker glial fibrillary acidic protein (GFAP) [41]. In another TUNEL assay, this time on the fetal and neonatal human brain, the authors, based on the lack of positive nuclei in the EGL, Purkinje cell layer and dentate nucleus, concluded that apoptosis mainly affected the glial cells [55]. More recent studies have rather suggested that active caspase-3 might be important for the differentiation of the Bergmann glia [122].

## 4. Non-Apoptotic Role of Caspase-3 in Cerebellar Development

In the normal developing and mature brain, caspases have been implicated in the regulation of a series of events that are functionally very relevant, such as selective axonal degeneration (axon pruning) and synapse weakening and elimination [123,124,125,126]. In cerebellum, it has long ago been suggested that caspase-3 might intervene directly in the proliferation and differentiation of the CGCs within the EGL [79]. To our knowledge, however, this matter has not been investigated further. Notably, when we recently analyzed the levels of activity of caspase-3 in cerebellar slices [78], we observed that the protease displayed low levels of activity in a subset CGCs, that these neurons were resistant to challenge with apoptosis inducers or oxidative stressors, and that it was possible to capture the existence of spots of intense caspase-3 activity in their processes, at a developmental stage when cortical synapses are formed in the cerebellum.

## 5. Future Perspectives

The study of postnatal cerebellar development offers unique opportunities to inspect NOND in vivo, as the process affects very high numbers of cells in a quite restricted temporal frame. This allows one to analyze the cellular and molecular mechanisms of regulation despite that apoptotic cells are very rapidly cleared from tissue. There are two main streams that, according to our opinion, should be pursued in future studies. First, it will be important to compare cerebellar development in altricial and precocial animals, as the latter have been, for the most, neglected in current studies. This will hopefully help to more deeply dissect the pathways governing NOND in relation to normal development. Second, it will be important to better analyze the role of caspase-3 in cerebellar NOLND, primarily taking into consideration the recent findings indicating that the protease may be also an important regulator of synaptogenesis.

## Figures and Tables

**Figure 1 ijms-19-03999-f001:**
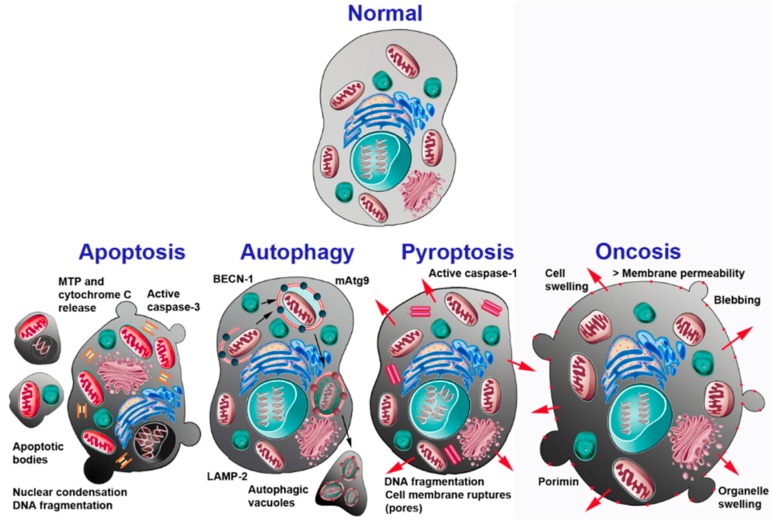
Schematic drawing of the main different types of cell death occurring in neurons. The main cellular and molecular events associated with each death process are indicated. Abbreviations: BECN-1 = beclin-1; LAMP-2 = lysosome-associated membrane protein 2; mAtg9 = mammalian autophagy-related protein 9; MTP = mitochondrial transition pore. Reprinted with permission from [2].

**Figure 2 ijms-19-03999-f002:**
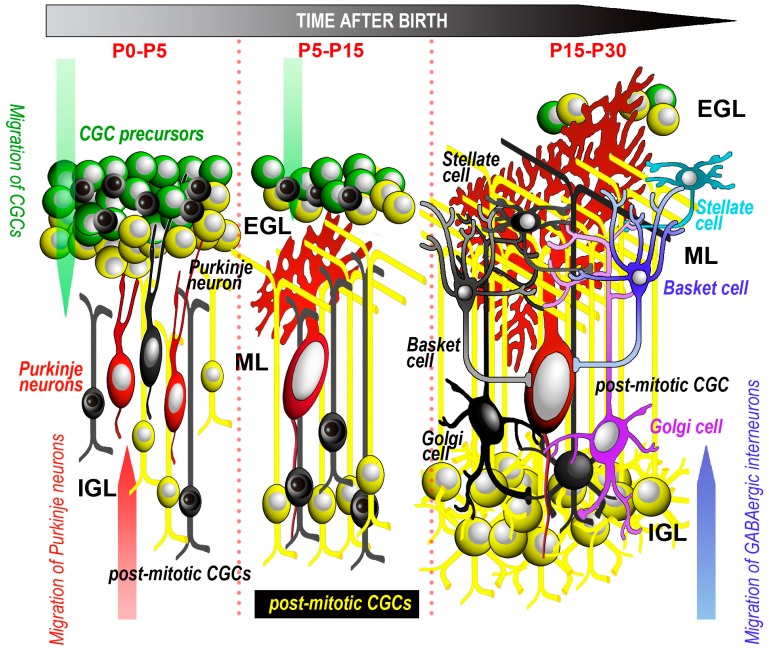
Schematic drawing of the main events related to cell proliferation, migration and apoptosis of cerebellar cortical neurons. Dead neurons are black. For simplicity, glial cells are not represented. Timetable refers to rodents and rabbits. See text for further explanations. Abbreviations: CGC = cerebellar granule cells; EGL = external granular layer; IGL = internal granular layer; ML = molecular layer; P = postnatal day.

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
