# Peer review of "Caspase-3 Mediated Cell Death in the Normal Development of the Mammalian Cerebellum"

_ijms, 2018, doi:10.3390/ijms19123999_

Reviewer 1 Report

In this manuscript, the authors chronologically review the mechanisms of naturally occurring neuronal death during cerebellar development focusing on caspase-3. I think it is intriguing and well organized, and worth publishing in International Journal of Molecular Sciences after minor revision.

[specific comments]

#1. (page 7, lines 244 and 245) “intrinsic pathway” and “extrinsic pathway” are oppositely described.

#2. (page 11, section 3.2.2.2.1.1.) References are missing in the first and second sentences.

#3. (page 16, line 588) The reference should be #78 (not #79).

#4. (page 17, line 611) as well as (as is missing).

#5. (page 17, 3.2.2.4.) Since the role of AIF in caspase-independent apoptosis is mentioned here, I think it should be described in detail in section 3.1.1.

#6. (page 19, line 722) “form the VZ” should be “from the VZ”, and the reference should be #120 (not #102).

#7. (page 20, line 755) The reference should be #78 (not #83).

Author Response

We wish to thank the reviewer for the appreciation of our work and for having spotted out all mistakes and inaccuracies.

In response we have amended them all and added a brief description of the role of AIF in apoptotic cell death (now at lines 253-258 of the revised MS).

Reviewer 2 Report

In the review manuscript by Lossi et al., the authors comprehensively review the past history of naturally occurring neuronal death and current scientific progress of caspase-3-mediated cell death in the normal development of the mammalian cerebellum. Although slightly lengthy, the manuscript is clearly and fluently written. Therefore, I would like to recommend this review article for the publication IJMS after the consideration of following minor points.

MINOR POINTS

Page 2, Line 41

The authors describe that caspase-independent programmed cell death consists of “cathepsin-dependent necroptosis”. However, recent scientific advances have identified that necroptosis is dependent on RIPK1-RIPK3-MLKL signaling pathway. Rather, cathepsins limit macrophage necroptosis through cleavage of RIP1, as reported by McComb et al. (J. Immunol. 192:5671-5678, 2014). Although necroptosis is not the main topic of this review, this point should be corrected. Also, Ref.8 is not directly related to necroptosis. The paper by Yamada et al. (Am J Pathol 186: 2798-2802, 2016) may be a better choice.

Page 7, Lines 244-245

The terms “intrinsic” and “extrinsic” are used oppositely.

Page 7, Line 249

“Cytochrome” should be specifically “cytochrome c”.

Page 10, Line 332 and Page 13, Line 479

The term “CPP32” is no longer used and should be avoided. Please use “Caspase-3”.

Page 10, Line 331 and page 14, Line 490

The term “interleukin-1β-converting enzyme” and “ICE” are no longer used and should be avoided. Please use “Caspase-1”.

In the section 3.2.2.2.3, the authors describe the phenotype of Bax-/- mice. However, it is currently believed that both Bax and Bak (and in some cases, Bok) work together to induce intrinsic apoptosis. To formally discuss the relationship between NOND of CGCs and intrinsic apoptosis pathway, it would be necessary to refer to the papers analyzing the phenotype of Bax-/-;Bak-/- double knockout mice.

Page 16, Line 605

“Tropomyosin kinase” should be “tropomyosin receptor kinase”.

I think it would be better to end this manuscript with a concluding section such as “conclusion”, “perspective on the future”, or anything else.

Author Response

We warmly thank the reviewer for her/his warm appraisal of our work.

We have coped will all suggestions and amended all mistakes/typos in the reviewed MS.

Specifically:

We have mentioned the role of RIPK1-RIPK3-MLKL signalingin necroptosis and changed ref. 8 as required.

We have amended section 3.2.2.2.3 as suggested and briefly discussed the phenotype of Bax-/-;Bak-/- double knockout mice (at lines 543-548 of revised MS).

We added a section on future perspectives as suggested (at lines 772-782 of revised MS).